# Research on supporting mechanism of ancillary service of PV system to grid energy efficiency based on multi-time and space-time operation

Cui Yong[1], Mingzhen Shao[2]*, Zhou Xiaoqian[3], Liu Wen[4], Ji Desen[5], Thomas Stephen Ramsey[6]

1 School of Economics and Management, Hubei Polytechnic University, Huangshi, China, 2 School of Economics, Henan University, Kaifeng, China, 3 School of Electric Power, Shanghai Dianji University, Shanghai, China, 4 Department of Engineering, University of Cambridge, Cambridge, United Kingdom, 5 Power Supply Service Center of State Grid Jiangxi Electric Power Company, Nanchang, China, 6 School of foreign languages, Three Gorges University, Yichang, China

* samezn@163.com

**Data Availability Statement:** All relevant data are within the paper and its Supporting Information files.

## Abstract

Under the background of high-energy penetration of new energy into the power grid, this paper takes the ancillary service capability of photovoltaic energy integrated into the grid as the starting point and builds a photovoltaic system reactive power service impact evaluation model on the grid energy efficiency. This is based on the multi-temporal and spatial scale operation mode, in order to study the supporting principles of photovoltaic system reactive power services on the energy efficiency of grid operation and the law of influence on system energy efficiency changes. In this way, the space for power system energy efficiency improvement and the reactive power service market value of renewable energy are explored to improve the renewable energy auxiliary services participation in the theoretical system of electric power spot market transactions. The research conclusions can provide a decision-making reference for system dynamic energy efficiency management and can assist relevant market entities to make optimal decisions in spot market transactions, and provide empirical data for improving the theory of renewable energy participation in auxiliary service market transactions.

## Introduction

Under the background of high-energy penetration of new energy into the power grid, the impact of new energy on the safe operation of the power grid is one of the hot topics today. However, the research on the mechanism and effect of reactive power support capacity on the energy efficiency of grid operation lacks corresponding attention. Mostly, power grid companies do not have a full understanding of the relationship between reactive power support capacity and grid energy efficiency. Furthermore, the reactive power support function of the

**Funding:** This study was funded by Talent Introduction Project Fund of Hubei Polytechnic University in 2022, and the project fund number is Humanities No. 5.

**Competing interests:** The authors have declared that no competing interests exist.

photovoltaic power generation system has not been fully tapped. The auxiliary service of photovoltaic power generation system is mainly to make full use of the residual capacity of the inverters in the photovoltaic power generation system and the SVG devices to provide reactive power for the power system, to realize the balance of reactive power of the power system and support for the voltage stability of the power system. The reactive power supporting equipment in the photovoltaic power generation system mainly includes inverters and SVG devices. When determining the plan for connecting the photovoltaic power generation system integrated to the power grid, there is a lack of effective methods to evaluate the impact of its reactive power support capacity on changes in the energy efficiency of grid operation. Therefore, the determined planning objectives for the photovoltaic power generation system integrated into the grid have a restrictive effect on the energy efficiency improvement in the grid company's later grid operations. The real-time balance of reactive power plays an important role in stabilizing market transaction electricity prices. Renewable energy represented by photovoltaic power generation is widely integrated into the power grid, researches on economic operation control, power quality, energy efficiency evaluation, optimal utilization of reactive power resources, energy-saving and emission reduction, and their impact on grid stability after grid connection have achieved certain results. However, there is a lack of mechanism explanation for the degree of its impact on the energy efficiency of the power grid. Clarifying the relationship between the reactive power resources in the renewable energy power generation and the energy efficiency of the power grid operation can further promote room for improvement in the energy efficiency management of the power grid. Under the multi-temporal and spatial scales operation mode of the system, based on the supporting capability of auxiliary services of renewable energy, how market entities in the power system can make effective market trading strategies for changes in power system energy efficiency to improve its income is an urgent problem for all entities to solve. Especially under the background that renewable energy has fully penetrated the power grid and the energy utilization structure has been transformed, it still needs theoretical basic research to mine the value of the reactive power resources of renewable energy based on the perspective of energy efficiency management to realize the optimal allocation of system resources and energy-saving and emission reduction of the power system.

## Literature review

At present, academic scholars mainly focus on the following four aspects concerning the energy efficiency of the power grid, the reactive service capability of the photovoltaic power generation system, the new energy grid-connected operation control, and its optimized scheduling.

Firstly, some research focuses on the reactive power support capability of inverters and SVG devices in photovoltaic power system and their coordination and optimization. Research [1, 2] adopted a maximum power point tracking (MPPT) technology with an improved perturbation and observation method (P&O) and it was applied to grid-connected photovoltaic systems. This research can realize that the photovoltaic system inverter had sufficient residual capacity of reactive power and a reasonable adjustment range. It laid a foundation for ensuring the operational stability of grid-connected photovoltaic systems. Research [3, 4] proposed a reactive power and voltage control strategy based on Monte Carlo (MC) method and robust optimization (RO) algorithm, and this strategy can reduce energy loss and voltage deviation to improve the operation control capability of the distribution network and the consumption capacity of the photovoltaic power generation system. Research [5–7] considered the diversity of the topology changes of the distribution network, reactive power and voltage partition

coordination control strategy based on photovoltaic system inverters was proposed, the strategy can effectively and timely control the voltage of the over-limit point within a safe range, and improve the stability of the node voltage of the distribution network. The above research mainly focuses on the application or improvement of reactive power optimization control technology in the photovoltaic power system. However, there is no corresponding research on the effect of reactive power support on-grid energy efficiency for the coordinated optimization of the inverter and SVG devices in photovoltaic systems.

Secondly, some research focuses on comprehensive benefit and energy efficiency evaluation of photovoltaic power systems. Research [8–10] proposed a real-time reactive power and voltage control method for active distribution networks that combined central and local reactive power and voltage to reduce voltage over-limits, improved system stability as well as minimized distribution network operating loss. Research [11] studied the calculation and evaluation methods of energy efficiency in photovoltaic power plants and found that photovoltaic modules, the conversion efficiency of inverters and system loss were the main factors affecting the energy efficiency of photovoltaic power generation; Research [12] designed a robust correlation index system for energy saving and loss reduction of the distribution network. It verified the guiding role of this energy-saving evaluation method during loss reduction of the distribution network through practical cases. The above research mainly focuses on the study of energy efficiency, benefits, and evaluation methods of the photovoltaic power generation system. However, there is limited research concerning the impact of the photovoltaic power system on power grid energy efficiency.

Thirdly, the research focuses on the efficient utilization, and carbon emission reduction of renewable energy power generation after renewable energy is connected to the power grid. Research [13] introduced carbon emission and carbon trading mechanism into the integrated energy system (IES) with distributed generation, gave a joint configuration method of power-to-gas (P2G) equipment and photovoltaic capacity as well as established the joint optimization model of P2G equipment and photovoltaic system. Research [14] considered the low-carbon operation of the IES and built a carbon capture and carbon storage equipment model, which effectively reduced the carbon emissions of the industrial park and promoted a high proportion of new energy consumption. The above research mainly focuses on the overall effect of grid-connected renewable energy on energy saving and emission reduction of the power system, as well as the relationship between environmental protection and power planning. However, the low-carbon development and energy-saving planning issues after the integration of renewable energy into the power grid did not adequately consider the improvement of energy efficiency in the later operation of the power system.

Fourth, some research focuses on the power quality of the power system after renewable energy is connected to the grid. Research [15] proposed an integrated control scheme of the energy management system (EMS) and renewable energy auxiliary service scheduling, used photovoltaic power generation system information monitoring gateway to collect renewable energy system operation data to support EMS for renewable energy power generation scheduling control. And considered that using the residual capacity of renewable energy to implement reactive power auxiliary services to improve system voltage quality and achieve the operation stability of large scale power systems in the background of high penetration of renewable energy. Research [16] summarized the currently commonly used total harmonic distortion (THD) reduction methods, and based on the mechanism of the adaptive filter and its application effect on harmonic suppression, future application development was proposed. Reference [17] proposed an adaptive predictive filtering method to reduce THD by adopting the least mean square (LMS), normalized LMS (NLMS) and leaky LMS (LLMS) algorithms, and the algorithm tests were implemented based on the single-phase standalone photovoltaic power

generation model to verify its validity. The simulation results indicate that the proposed method can reduce the THD of photovoltaic systems by more than 70%. The new energy grid-connected inverter technology has improved rapidly and can effectively control harmonics. Research [18] pointed out that a dynamic reactive power compensation device should be installed at the harmonic source to compensate for the reactive power demand of rapidly changing loads, improve power factor, filter out system harmonics and reduce the injection of harmonic currents into the power system. Research [19] proposed a unified control strategy of photovoltaic grid-connected and active filters, which can realize photovoltaic grid-connected power generation as well as realize harmonic compensation and improve system utilization. Research [20] pointed out that adopt full-bridge inverter circuit and high-frequency switching electronic devices in inverters can make the output harmonics have been well controlled. And the proposed method of space transformation can precisely control the harmonics injected into the grid current so that the low-order harmonic content of the output current of the photovoltaic inverter is the smallest under the optimal switching times. The above research mainly focuses on the problem of harmonic suppression after renewable energy is connected to the grid. With the advancement of photovoltaic inverter production technology, the impact of harmonic current in photovoltaic power generation systems on all aspects of the grid will be further reduced. The differences between relevant literature and this research are shown in Table 1 below.

Based on the research progress and shortcomings of existing literature, this paper may have the following academic contributions and values:

1. By analyzing the boundary operating characteristics of the photovoltaic power generation system after it is integrated into the grid, typical operation modes of power systems, typical operation load rates, different grid structures, and two boundary reactive power operation modes of inverter devices or SVG devices in photovoltaic power generation systems are simulated. The correlation effect analysis model of photovoltaic power system reactive power service to support grid energy efficiency support, which is based on Pearson method and power system voltage deviation, active loss, and line loss rate as the core indicators are designed. Taking the auxiliary service capability of the photovoltaic power generation system as the starting point, the mechanism of the reactive power service capability of the photovoltaic power generation system support the energy efficiency of the power grid operation is studied, and the effect of the reactive power supporting capability of the photovoltaic power generation system on the energy efficiency supporting of the power grid operation is planned to be verified. It provides decision-making reference for the planning and design, investment cost control, and energy efficiency management of the grid company before the photovoltaic power generation system is integrated into the power grid.

2. In this paper, a mean-semi-absolute deviation and t-test system operation energy efficiency correlation effect evaluation model is constructed. Through calculating the deviation of voltage or energy consumption, the difference in the impact of the reactive service capability of inverters or SVG devices in the photovoltaic power generation systems on changes in the energy efficiency of grid operations is verified. Based on the comparison of different combined operation modes, the effect of reactive power support capacity of photovoltaic power generation system on energy efficiency support of grid operation is verified. It is expected to provide decision-making references for planning and design, investment cost control, and energy efficiency management of grid companies before and after the photovoltaic power generation system is integrated into the grid. Meanwhile, taking the supporting effect of the reactive power service of the

**Table 1. Comparison table of differences between the research in this article and the main related literature research.**

| Time | Research | Research content | Research solves the major problem | Research goal | Research method |
|------|----------|------------------|-----------------------------------|---------------|-----------------|
| 2021 | [8] | Optimal local voltage control strategy for photovoltaic inverters in active distribution networks | The problem of voltage exceeding the limit in the process of continuous increase of the penetration rate of photovoltaic power generation in the active distribution network. | Reduction of voltage overruns and minimization of power network losses as the goal | Convolution Neural Networks (CNN) |
| 2021 | [9] | A voltage adjustment method for active distribution network considering reactive power optimization of substations | Fluctuations in photovoltaic power generation lead to significant voltage fluctuations in active distribution grids | Running loss minimization as the goal | Non-dominated Sorting Genetic Algorithm-II (NSGA-II) |
| 2021 | [10] | Reactive power optimization algorithm of distribution network with the photovoltaic power generation system | After the photovoltaic power generation system is connected to the distribution network, the power quality decreases, the system's active power loss increases, and the system is unstable. | System stability and the economy as a goal | Non-dominated Sorting Genetic Algorithm-III (NSGA-III) |
| 2021 | [13] | Integrated energy system carbon emissions and carbon trading, as well as power-to-gas equipment and photovoltaic co-optimized configuration | How to improve the utilization efficiency of IES and the supporting capability of reasonable equipment configuration in IES for carbon neutrality | The goal is to minimize the total cost of system operation | K-means |
| 2020 | [7] | Voltage Coordinated Control Strategy of Distribution Network with Distributed Photovoltaic Generation | Distributed photovoltaic grid connection brings problems such as many centralized control dimensions and complex control process of voltage to the distribution network. | Realize fast control in case of system emergency and reach the voltage safety range | Clustering By Fast Search and Find Of Density Peaks (CFSFDP) |
| 2019 | [3] | Three-stage reactive power and voltage control strategy of inverter based on high-penetration photovoltaic power generation system | Uncertainty of PV power output and load demand | reducing energy loss and voltage deviation as a goal | Robust Optimization (RO) |
| 2019 | [4] | Simulation of consumption capacity of distribution network and research on voltage control strategy under high penetration rate of photovoltaic power generation | The imbalance between photovoltaic output and load leads to reverse the tide in the line current, causing voltage out-of-limit problem | improving photovoltaic consumption capacity as a goal | Monte Carlo method (MC) |
| - | This research | Research on the supporting mechanism of photovoltaic system auxiliary service to grid energy efficiency | Mining of auxiliary service resource value of photovoltaic power generation system and its impact on the dynamic energy efficiency of the power grid | The correlation between the reactive power service of the photovoltaic system and the energy efficiency of the grid, and the law of dynamic changes in the energy efficiency of the system | Pearson correlation coefficient, mean-semi-absolute dispersion, and t-test |

photovoltaic power generation system on the grid operation energy efficiency as an example, it is intended to reveal the general law of the impact of renewable energy on the grid operation energy efficiency after grid connection, to prove that renewable energy has participated in the auxiliary services of the power system to the system, to prove the extent of renewable energy's impact on the energy efficiency of the system after participating in auxiliary services of the power system. It provides a theoretical basis for the construction of a market mechanism for auxiliary services containing renewable energy and transaction optimization decisions.

3. The research on the supporting mechanism of a single photovoltaic power generation system to the grid energy efficiency and the dynamic energy efficiency analysis method of multi-temporal and spatial scales can be further applied to the dynamic energy efficiency analysis in the grid cluster mode such as photovoltaic cluster or wind and solar cluster. Through the coupling mechanism and division of different cluster modes research realize

the power grid lean energy efficiency management, and provide ideas and methods for power grid energy conservation and loss reduction.

## Analysis of the impact of photovoltaic power generation system on-grid energy efficiency

The reactive power demand of photovoltaic power plants mainly originates from the transformer, collector line, and delivery line. The active power output of photovoltaic power plants fluctuates and is random. At the same time, the reactive power demand for photovoltaic power plants will change with the actual power output of photovoltaic power plants. The SVG device in a photovoltaic system is connected in parallel to the bus of low voltage side of the main transformers, which mainly compensates the reactive power requirement of main transformer and transmission line, and SVG generates the maximum reactive power to support the grid voltage in case of power grid failure. Based on the reactive power regulation capability of the inverter itself (i.e., when the active power of the inverter is at full load, i.e., 1 pu (per unit), its reactive power output can be adjusted between—0.484 pu and + 0.484 pu), the reactive power demand of transformer and collector can be compensated locally. Due to the different capacities of the photovoltaic power system connected to the power grid, the photovoltaic power system has an impact on the node voltage and active power loss.

On the other hand, after the photovoltaic power generation system is connected to the grid, a high-order harmonic current will be generated, and the harmonic current will have a certain impact on the energy consumption of the grid. Research [21] is based on the energy efficiency evaluation system of the distributed generation connected to the power grid. The multi-dimensional function partial differential calculation method is used to calculate the contribution of a single evaluation index to the energy efficiency level of the entire distribution system in a fixed state. The research results indicate that the contribution of the harmonic distortion rate of current to the energy efficiency of the power grid is the smallest, which is only 0.448, the contribution of the length of the overhead line is the second, which is 1.714, and the contribution of the distributed capacity is the largest, which is 5.479. At the same time, according to China's GB/T 33593–2017 "Technical Requirements for Distributed Power Grid-connected", GB/T 31464–2015 "Grid Operation Guidelines" and GB/T 14549–1993 "Power Quality Harmonics of Public Power Grids", the harmonics are proposed requirements. It is required that the grid-connected inverters themselves have the function of harmonic control, and the harmonic distortion rate of the grid-connected inverters does not exceed 5%. Therefore, the photovoltaic power generation system with grid connection conditions has little impact on the energy efficiency of the power system after grid connection. To sum up, although the power grid loss is related to the harmonic distortion rate of current, this paper mainly studies the supporting mechanism of a single photovoltaic system on the power grid energy efficiency, and its impact can be ignored. In order to quantitatively analyze the energy efficiency of the photovoltaic power system connected to the power grid, this paper selects voltage deviation, active power loss, and line loss rate of each node in the power grid as the primary basis for comprehensive energy efficiency evaluation.

## Model of energy efficiency correlation analysis

### A. Energy efficiency model of power grid operation

**1) Constraints on the stable operation of the power grid.**   Balanced regulation of active and reactive power of each node in the system under steady-state operation should be mainly considered during power grid energy efficiency management, and corresponding stability

requirements should be met to ensure the safe operation of the power grid.

$$
s.t. \begin{cases} P_{Gi} - P_{Li} - U_i \sum_{j=1}^{N} U_j (G_{ij} \cos\theta_{ij} + B_{ij} \sin\theta_{ij}) = 0 \\\\ Q_{Gi} - Q_{Li} - U_i \sum_{j=1}^{N} U_j (G_{ij} \sin\theta_{ij} + B_{ij} \cos\theta_{ij}) = 0 \\\\ P_{Gi}^{\min} \leq P_{Gi} \leq P_{Gi}^{\max} \\\\ Q_{Gi}^{\min} \leq Q_{Gi} \leq Q_{Gi}^{\max} \\\\ U_i^{\min} \leq U_i \leq U_i^{\max} \\\\ I_i^{\min} \leq I_i \leq I_i^{\max} \\\\ \omega^T l = 1 \end{cases}
\tag{1}
$$

In the process, N is the number of nodes: $Q_{GI}$, $Q_{Li}$ is the generator and load reactive power, kVar; $U_i$ is the voltage amplitude of node i, kV; $\theta_{ij}$ is the phase angle difference between two nodes, rad; $G_{ij}$, $Bi_j$ are the real and imaginary parts of node admittance, S. $\omega^1$ is the probabilities of different operation modes, $\iota$ is the all column 1 vector.

**2) Calculation equation of power flow in the system (active power and reactive power).**  The Newton-Raphson method is used to calculate the power flow of the power grid. The Newton-Raphson modified equation is as follows:

$$
[\Delta P \ \Delta Q]^T = J[\Delta\theta \ \Delta U]^T
\tag{2}
$$

In the formula, $\Delta P$ is the active power unbalance; $\Delta Q$ is the reactive power unbalance; $\Delta\theta$ is the voltage angle balance; $\Delta U$ is the amplitude correction; J is the Jacobian matrix.

**3) Voltage deviation index.**  The voltage deviation formula is as follows:

$$
\Delta U = \frac{(U_i - U_s)}{U_s} \times 100\% \quad i = 1, \ldots, N
\tag{3}
$$

In the formula, $U_i$ is the node voltage, kV; $U_s$ is the system nominal voltage, kV. Under normal operation mode, the absolute sum of positive and negative deviations of supply voltage does not exceed 10% of nominal system voltage.

**4) Index of active power loss in the power grid.**  The nodal equivalent power method is used to calculate the active power loss. The formula is as follows:

$$
P_{loss} = \frac{P^2 + Q^2}{U^2} Rt
\tag{4}
$$

In the formula, R is the branch resistance, $\Omega$; t is time, s; P is the node active power, kW; Q is the node reactive power, kVar.

**5) Comprehensive energy efficiency calculation of the system.**  The comprehensive energy efficiency of the system is calculated by subtracting the statistical line loss rate by 100%.

The formula for calculating the statistical line loss rate is as follows:

$$\lambda = \frac{A_{\text{loss}}}{A_{\text{g}}} \times 100\% \tag{5}$$

In the formula, $A_{loss}$ is the line loss power, kW·h; Ag is the power supply, kW·h.

## B. Correlation model of reactive power support and energy efficiency of the power grid in photovoltaic system

**1) Correlation coefficient.** Pearson correlation coefficient is used to calculate the correlation. The formula is as follows:

$$r = \frac{\sum_{i=1}^{n}(x_i - \bar{x})(y_i - \bar{y})}{\sqrt{\sum_{i=1}^{n}(x_i - \bar{x})^2}\sqrt{\sum_{i=1}^{n}(y_i - \bar{y})^2}} \tag{6}$$

In the formula, $r$ is the Pearson correlation coefficient, $|r| \leq 1$; x,y are the sample variables, $\bar{x}, \bar{y}$ is the sample mean, n is the sample number, i is the sample number.

The correlation coefficient can describe the degree of correlation between variables and the positive and negative attributes. The positive and negative values of r represent the positive and negative correlation of variables. $|r|$ represents the degree of correlation of variables. The larger $|r|$ is, the stronger the correlation degree is, and the smaller $|r|$ is, the weaker the correlation degree is.

**2) T-test and improved mean-semi-absolute deviation model.** Mean-semi-absolute deviation [22, 23] is used to evaluate the degree of deviation of voltage deviation and active power loss from the mean under different operation modes. The improved mean-semi-absolute deviation formula is as follows:

$$f = \begin{cases} p - \bar{p} & p > \bar{p} \\ 0 & p \leq \bar{p} \end{cases} \tag{7}$$

In the formula, p is the simulation calculation values of voltage and energy consumption, $\bar{p}$ is the voltage deviation and average energy consumption. The mean value of voltage deviation is the sum of all data of voltage deviation divided by the total number of voltage deviation data; the mean value of energy consumption is the sum of all data of energy consumption divided by the total number of energy consumption data.

The test is used to test the hypothesis of the deviation of voltage deviation and active power loss under different operation modes to verify the difference of positive and negative correlation between reactive power support capability of the photovoltaic power system and power grid operation efficiency under different combined operation modes.

$$t = \frac{\bar{X} - \mu}{S/\sqrt{n}} \tag{8}$$

In the formula, $\bar{x}$ is the sample mean, μ is the expectation, S is the standard deviation, n is the sample size. When $|t| \geq t_{a/2}$, the original hypothesis is negated. Otherwise, the original hypothesis is accepted.

## C. Conditions of simulation assumption calculation

There are 7 main conditions as follows:

1. In the future, the popularization and utilization of reactive power optimization system software for photovoltaic power generation systems will be more extensive, that is, it is assumed that the reactive power optimization control system in photovoltaic power stations can realize the coordination of reactive power support functions of SVG or inverter devices.

2. It is assumed that the harmonic current generated by the grid connection of a single photovoltaic power generation system has a negligible impact on the energy consumption of the grid.

3. It assumes that the load characteristics of a randomly selected system are specific, the peak-valley difference rate (percentage of the difference between maximum load and minimum load divided by maximum load) corresponding to the load characteristic curve is not affected by load rate (percentage of the average load to the maximum load ratio in the specified time) of the operating system. It also assumes three different load rates (see Table 2) are analyzed and calculated with various combinations, namely winter large-scale operation mode (all components in the system are put into operation), summer large-scale operation mode and summer small-scale operation mode (in which the units with the least input and the highest economic benefit are selected according to a minimum load of the system displayed in the system for a long time). In the future, the three operation modes will be referred to as Large-scale Winter, Large-scale Summer, and Small-scale Summer respectively;

4. It assumes that photovoltaic power system is connected to the 14-node system (a grid structure in the International Power System Database) of the IEEE (Institute of Electrical and Electronics Engineers), and it simulates photovoltaic power system connected to the IEEE 57-node system for further theoretical verification;

5. In different node systems, the position of the photovoltaic power generation system incorporated into the node is different, and the impact on the energy efficiency of the power system is different. This article assumes that the merged node position is randomly selected. For two (IEEE57, IEEE14) node systems, choose to merge from node 7 respectively.

6. It is assumed that the fundamental operation data of IEEE14 and IEEE57 bus systems are used to calculate corresponding energy efficiency indexes as the initial state for analysis and comparison (i.e. the state without connection to photovoltaic power system), hereinafter referred to as the initial state;

7. This paper assumes that under different combinations of operation modes, there is a difference in terms of the correlation between voltage deviation and power loss of inverters or SVG devices with reactive power support capability in the photovoltaic power system. When the original hypothesis is negated. Otherwise, the original hypothesis is accepted. The test level is generally less than 0.05, indicating that the probability does not exceed 0.05.

**Table 2. Load rate value of the system.**

| Operation mode | Large operation in summer | Large operation in winter | Small operation in summer |
|---|---|---|---|
| Load rate | 60% | 40% | 22% |

## Case study

The total installed capacity of a photovoltaic power plant is 60 MW with six collector lines in total. Each collector line is connected with 10 groups of PVGU (Photovoltaic Generation Unit) in series, and each group of PVGU has a capacity of 1 MW, which is connected to collector lines of 10 kV through step-up transformers. The capacity of the 110 kV main transformer is 63 MVA (capacity unit), and the length of transmission lines is 80 km. The LGJ-185 conductor (line model) is used. SVG devices can be directly parallel connected to a 10 kV bus in a photovoltaic booster plant. Based on the 110 kV high-voltage side, $S_B$ = 100MVA, and in order to simplify the calculation, it is set to be 1pu. The equivalent circuit of the photovoltaic power system connected to the power grid (due to the limited length of the article, the parameter derivation of the equivalent circuit is omitted) is shown in Fig 1.

With the goal of quantitative analysis and calculation, the system load ratio under three operation modes of Large-scale Winter, Large-scale Summer, and Small-scale Summer is assumed as shown in Table 2. At the same time, two critical operation modes are assumed: mode 1 shows that when output values of the active power of a photovoltaic power plant (indicating the necessary power to keep the equipment running) and reactive power of inverters reach the maximum, the reactive power output value of SVG devices is 0. Mode 2 indicates that when output values of the active power of a photovoltaic power plant and the reactive power of SVG devices reach the maximum, the reactive power output of inverters is 0. The combined operation mode of reactive power support of the photovoltaic system is shown in Table 3. In the table, the capacity configuration of SVG is 10 MVar (unit of reactive power capacity), i.e., 0.1 pu (rated capacity of equipment divided by reference capacity), the maximum value of reactive power output of SVG is 0.1 pu, the active power of photovoltaic is 60 MW, i.e., 0.6 pu, and the maximum output of reactive power of inverters is 48.4% (according to the technical parameters of inverters), i.e., 0.29 pu.

## D. Coefficient of system voltage deviation

According to formulas (1), (2), (3), and (6), the voltage deviation correlation between the combination mode in Table 3 and the initial value of voltage deviation calculated based on the IEEE14 bus system (initial state) is analyzed. The calculated results are shown in Fig 2. In Fig 2, the voltage deviation correlation coefficient under the combination mode of inverters or SVG reactive power output is 1. The correlation coefficients between the node voltage deviation of mode 1 or mode 2 and the initial value of voltage deviation of IEEE14 bus system are 0.6968 to 0.9995, respectively, showing a strong correlation. It shows that the effect of mode 1 and mode 2 on the system voltage deviation resembles, indicating that improvement of voltage

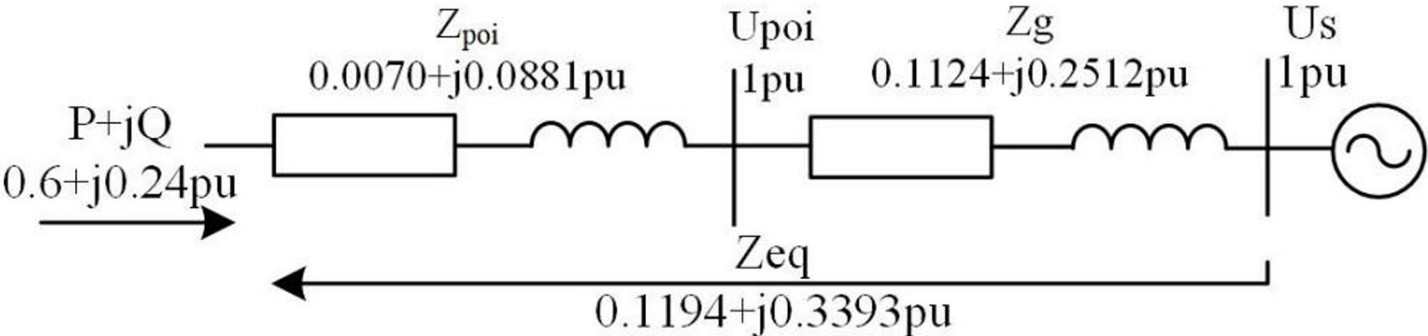

**Fig 1. Equivalent circuit diagram.**

**Table 3. Photovoltaic system reactive support combination unit: Pu.**

| Mode | Photovoltaic active power | Inverter reactive power | SVG reactive power |
|---|---|---|---|
| Mode 1 | 0.6 | 0.29 | 0 |
| Mode 2 | 0.6 | 0 | 0.1 |

deviation is positively correlated with the reactive power support capability of inverters or SVG devices, and through the comparative analysis of normalized load curves (as shown by the asterisk mark. It is to facilitate comparative analysis so that the normalized load curve can be directly put into the same graph. In a later analysis, the load curve is directly quoted, and repeated drawing is no longer needed). This study finds that the correlation coefficient of voltage deviation fluctuates with the change of load.

The correlation coefficients of voltage deviation are calculated based on the comparison of three load rates in Table 2 for Large-scale Summer, Large-scale Winter, Small-scale Summer, and initial states. As shown in Fig 3, the correlation coefficients are all bigger than 0.9, meaning that the load rate has a strong correlation with the impact on the power grid. The variation trend of correlation between operation modes of Large-scale Summer and Large-scale Winter, and those of Large-scale Winter and Small-scale Summer, are the same and are less affected by system load change (compared with the load curve in Fig 1). The correlation between operation modes of Large-scale Summer and Small-scale Summer, and the correlation index between voltage deviation and initial state voltage deviation under operation modes of Large-scale Summer, Large-scale Winter, and Small-scale Summer are greatly affected by system load change (compared with the load curve in Fig 1), and the correlation coefficient also fluctuates greatly.

## E. The correlation coefficient of system active energy consumption

Active power is calculated according to formulas (1), (2), (4), and (6), and the initial active power loss of the IEEE14 system is 0.1318 pu. The variation of active power loss under

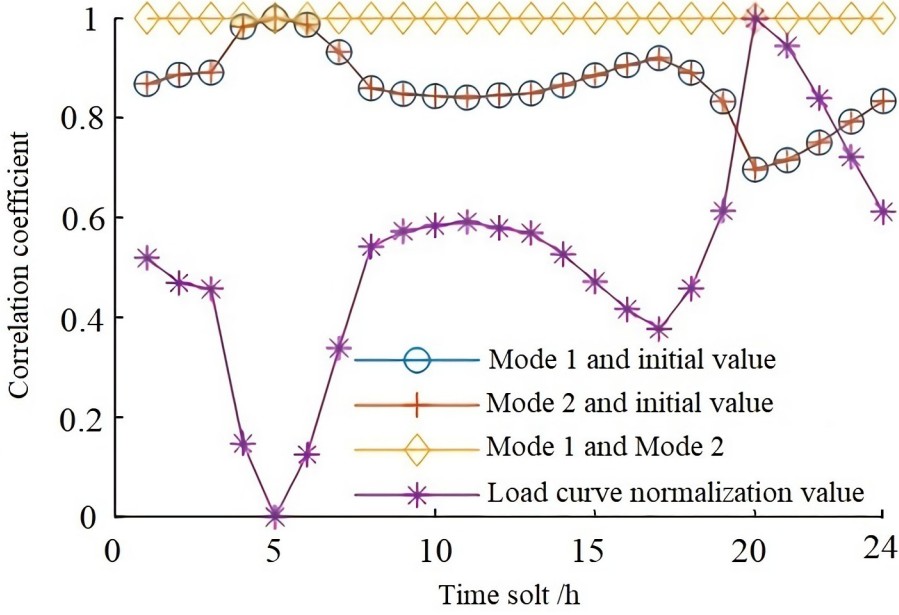

**Fig 2. Reactive power support combination method and initial voltage deviation correlation coefficient.**

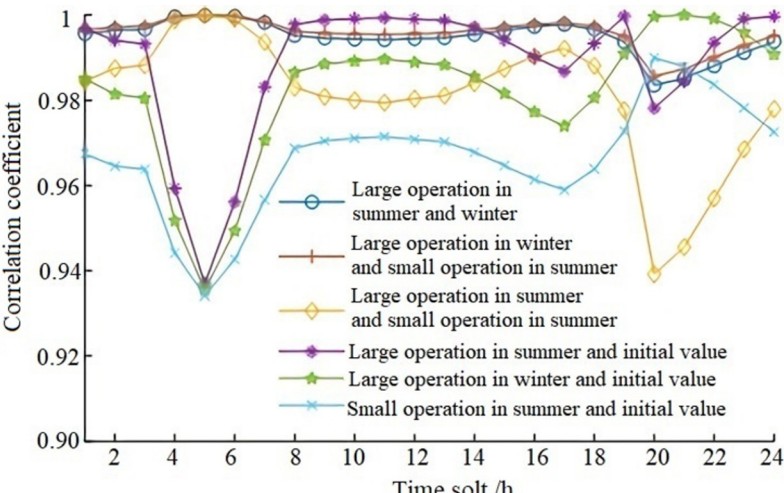

**Fig 3. The correlation coefficient of voltage deviation under multiple combined operation modes.**

different operation modes and different load rates is shown in Fig 4. The calculated active power loss under two modes in Table 3 is reduced to below the initial loss value during the low load period from 04:00 to 07:00. It can be seen from Fig 4 that after a photovoltaic power system is connected to the power grid, active power loss of the system is highly correlated with load rate under different operation modes.

The correlation coefficient of active power loss is shown in Table 4. Based on the three load rates in Table 2, the correlation coefficients of active power loss between mode 1 and mode 2 and between Large-scale Summer, Large-scale Winter, and Small-scale Summer are calculated; the correlation coefficient between grid loss of mode 1 and mode 2 is 1, and the correlation coefficient of active power loss under Large-scale Summer, Large-scale Winter, and Small-scale Summer modes are close to 1, which indicates that when the active power of a photovoltaic system is constant, inverters or SVG devices work separately, and the influence on active power loss tends to be the same under the operation modes of Large-scale Summer, Large-scale Winter, and Small-scale Summer.

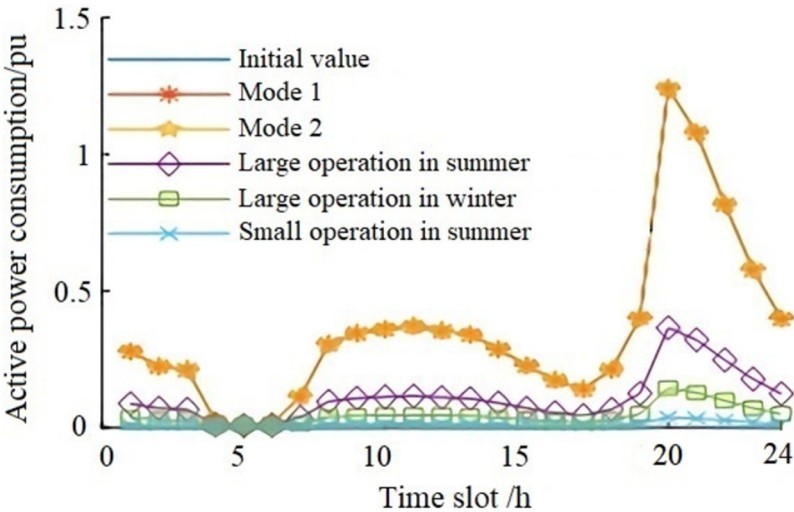

**Fig 4. Trend diagram of active power loss in the system.**

**Table 4. The correlation coefficient of active power loss under different operation modes.**

| Mode 1 and mode 2 | Large operation in summer and winter | Large operation in summer and a small operation in summer | Large operation in winter and a small operation in summer |
|---|---|---|---|
| 1 | 0.9998 | 0.9956 | 0.9972 |

## F. System comprehensive energy efficiency analysis

Based on the operation modes of Tables 1 and 2, and according to formulas (1), (2), (4), and (5), the comprehensive energy efficiency of the power grid after the photovoltaic connection is quantitatively analyzed by 100% subtracting statistical line loss rate. In order to make a comparative analysis, the initial energy efficiency of the system is calculated based on the IEEE14 bus system (according to the 24-hour load curve in Fig 1) and then compared with the energy efficiency of the system calculated according to three operation modes (Table 2) after it is connected to the photovoltaic power system (as shown in Fig 5).

From Fig 5, it can be seen that the energy efficiency of the power grid under each operation mode is higher than the initial value curve of energy efficiency. When the load is at the peak value between 20:00 and 23:00, the energy efficiency of the power grid decreases under mode 1 and mode 2, but it is still above the initial value curve of energy efficiency. Generally speaking, the energy efficiency of the system has been improved under each mode of operation.

## G. Analysis of alliance operation auxiliary service combination strategy

Based on the results of correlation analysis between reactive power support of the photovoltaic system and the energy efficiency index of power grid under different operation modes, the t-test is used to verify the difference of correlation effects. According to formula (8) and its hypothesis, the three operation modes of mode 1, mode 2, and Table 3 in Table 2 are combined for calculation and analysis respectively, and 12 t-test curves of voltage deviation and functional consumption under six combination modes can be obtained by simulation. Due to the limited length of the article, the minimum t-test value is selected for analysis (e.g., If the smallest t-test value can verify relevant results, then other cases can be proved as well). As can be seen from Fig 6, in the period other than 05:00–05:59, the t value of voltage deviation and the

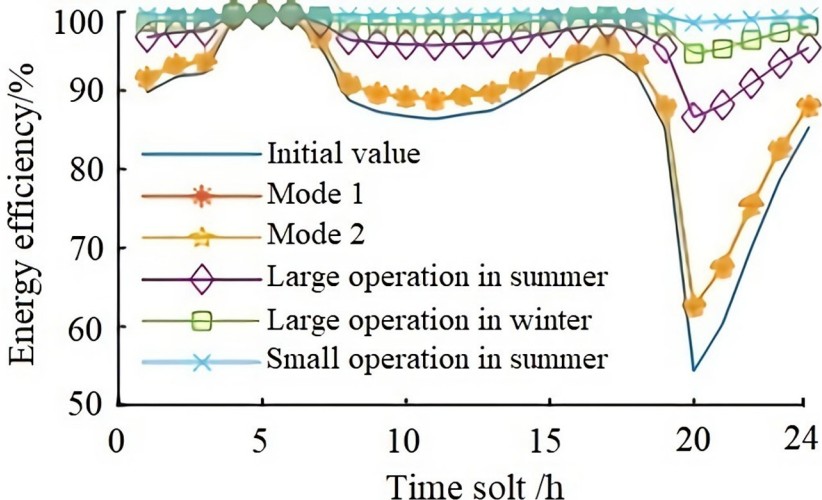

**Fig 5. The trend of system comprehensive energy efficiency.**

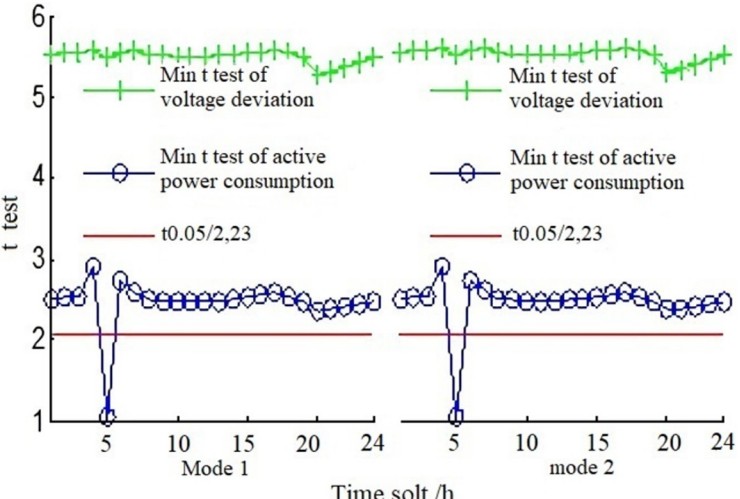

**Fig 6. T-test diagram of voltage deviation and active power consumption of IEEE14 node.**

active energy consumption is larger than the corresponding value of t0.05/2,23, namely 2.0687 (t0.05/2,23 means that the test level is 0.05 and the degree of freedom is 23). Therefore, according to formula (8), the original hypothesis can be negated (see no. 6 in 3.3 for original hypothesis), and it can be known that most of the time, there is no significant difference in the positive correlation between reactive power support of inverters or SVG devices, voltage deviation and active energy consumption improvement. According to the t values calculated under other operation combination modes, it can be seen that the positive correlation between the reactive power support of photovoltaic system and the energy efficiency improvement of the power grid is not significantly different from the improvement under different combinations of operation modes, that is, most of the time, the correlation is not affected by the combined operation mode.

## H. T-test of voltage deviation and energy consumption based on different grid structures

In order to further verify the difference of the influence of reactive power support of a photovoltaic system on the energy efficiency index of power grid under various combined operation modes, node 7 of the IEEE57 system is simulated to connect to the photovoltaic power system. The t-test values of voltage deviation and energy consumption are shown in Fig 7.

In mode 1 and mode 2 of Table 3, the t-test value of voltage deviation under operation modes of Large-scale Summer, Large-scale Winter, and Small-scale Summer is greater than t0.05/2,23, namely in the time period beyond 04:00 to 07:00. Under operation modes of Large-scale Summer, Large-scale Winter, and Small-scale Summer, the influence of reactive power support of inverters and SVG on voltage deviation and functional consumption does not differ. In the time period beyond 19:00–21:00, the t-test value of active energy consumption under three operation modes is greater than t0.05/2, 23, that is to say, in the time period beyond 19:00–21:00, the effect of reactive power support of inverters and SVG devices on all active energy consumption of Large-scale Summer, Large-scale Winter, and Small-scale Summer does not differ. In summary, there is no significant difference in the positive correlation between reactive power support of inverters and SVG devices, as well as voltage deviation and active energy consumption improvement most of the time. At the same time, the t values

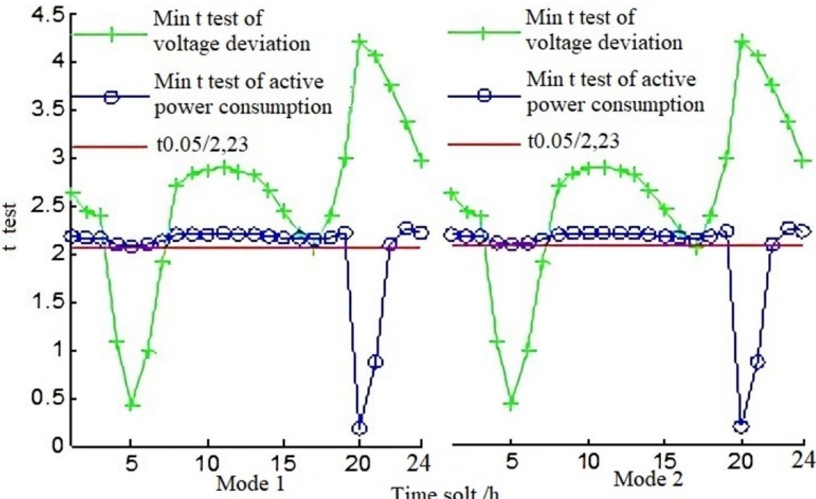

**Fig 7. T-test diagram of voltage deviation and active power consumption of IEEE57 node.**

calculated from other operation combinations show that the positive correlation between the reactive power support of photovoltaic system and the energy efficiency improvement of the power grid is not significantly different from the improvement under different combinations of operation modes, that is, most of the time, the correlation is not affected by the combined operation mode.

## I. A comprehensive discussion on the correlation between the photovoltaic system reactive power support and the grid energy efficiency

### (1) Comprehensive case analysis.

1. After the photovoltaic power generation system is integrated into the power grid, within 00:00–24:00, the correlation coefficient between the voltage deviation value and the initial state voltage deviation in a variety of operating states fluctuates between 0–1, and the fluctuation trend changes greatly. This generally shows a positive correlation indicating that it is greatly affected by changes in system load.

2. After the photovoltaic power generation system is integrated into the power grid, under different combinations of operation modes, the correlation coefficient indicators of active loss and voltage deviation are above 0.9 within 00:00–24:00, showing a positive correlation. Its correlation coefficient fluctuates with the change of the load rate, which shows that the active power loss and voltage deviation of the system are closely related to the change of the load rate.

3. When the active power of a photovoltaic power generation system is constant, within 00:00–24:00, the correlation coefficient of active loss in different operating modes is above 0.99, that is, the reactive power support capacity of the photovoltaic power generating system is basically not affected by different operating modes and it affects the power system. The trend of the effect of active power loss is consistent; at the same time, the overall energy efficiency change trend of the system can reflect that at the lowest point of the initial value of energy efficiency (at 20:00) and other times, the energy efficiency values of different operating modes are higher than the initial value of energy efficiency; At the same time, the comprehensive energy efficiency change trend of the system can reflect that at the lowest

point of the initial value of energy efficiency (at 20:00) and at other times, the energy efficiency values of different operating modes are higher than the initial energy efficiency value.

4. Based on the T-test effect in the statistics, that is, the analysis in H shows that there is no significant difference between the positive correlation of reactive power support capacity of the photovoltaic power generation system to the improvement of grid energy efficiency, and the improvement effect of the different operation combination modes, that is, the combination operation mode is not affected by most of the time.

5. The positions of the merged nodes of the photovoltaic system assumed in this article are randomly selected, and the selected node system is representative. When different nodes are merged, the energy efficiency change curve and correlation index are different. The access point based on the best energy efficiency can be determined by the system-wide node access simulation according to the analysis model constructed in this paper. In general, the integration of photovoltaic power generation systems from different nodes is effective in improving the energy efficiency of system operations.

6. Different types of renewable energy, under the assumptions in C, are simulated and verified based on different reactive power support boundary conditions, the system's multi-time scale load demand, and multiple operating modes. Therefore, similar results can be obtained.

**(2) Insufficiency of article research and prospects for future research.** The above results are based on six assumptions in C and the operation modes designed according to the operation characteristics of the power grid (Tables 2 and 3).

1. Due to the differences in reactive power equipment and reactive power support capabilities of other renewable energy sources, the changing trend and correlation index values of the system energy efficiency will be different from the results of this paper. This research provides an energy efficiency evaluation and analysis method based on the multi-temporal and spatial scale operation mode, however, in the process of selecting the location of renewable energy to be integrated into the grid, due to the constraints of power construction channels, investment costs, and other factors, the operating energy efficiency of the system may not be fully considered. Therefore, research based on multiple operating scenarios and constraints of planning and construction needs to be further expanded. At the same time, the application of this method needs to be combined with the actual operating data of the power grid to analyze and compare. It also provides data support for the market value of renewable energy reactive auxiliary services based on the perspective of system energy efficiency analysis and verifies the improvement effect of system operation efficiency.

2. In this paper, by analyzing the harmonic control function of the photovoltaic grid-connected inverters themselves and the contribution of the harmonic distortion rate of current of the photovoltaic power generation system to the energy efficiency of the power grid, the research case ignores the impact of harmonics generated by a single photovoltaic power station on the line loss of the power grid. Due to its small contribution, the photovoltaic power generation system has little influence on the changing trend of the grid energy efficiency; although the research method in this paper has laid a theoretical foundation for the dynamic energy efficiency analysis of the grid under the photovoltaic cluster mode, in the actual grid operation, the load is constantly changing where changes and differences in power load characteristics will affect the content of harmonic currents in the power system. Especially in the mode where the power system is connected to a large number of

photovoltaic clusters, the harmonic superposition or cancellation effect of different photovoltaic clusters after grid connection will cause an impact on the energy consumption and reliability of power grid operation. Therefore, the degree of influence of photovoltaic power generation clusters on the dynamic energy efficiency change trend of the power grid needs further research.

## Analysis of operating cost of a photovoltaic power plant based on system reactive power support

The SVG capacity of photovoltaic power systems is usually 20%-30% of the capacity of power plants. Assuming that 100 MW photovoltaic power plants are equipped with 20 MVar SVG devices, the SVG loss is set as 0.2% of the output capacity during system operation, and the average power cost is 0.0714 USD/kW·h, then the annual power consumption of 20 MVar SVG is 350,400 kW·h (20000Var*0.2%*24h*365days), and the operation loss will be 625,464 USD (350400·h*0.0714USD*25Years) after 25 years. Through research on the correlation between reactive power support of inverters and SVG devices and power grid, inverters can substitute SVG devices for their equivalent reactive power support. Therefore, the capacity configuration of SVG can be reduced during design. At present, the unit cost of 35 kV suspended SVG is about 14.2857USD/kVar. By reducing 10 MVar capacity allocation, the cost will be reduced to 142857USD, and the operating loss will also be reduced.

## Conclusion

This paper takes the reactive power capabilities of the photovoltaic power generation systems as the starting point to study the role of its reactive power capabilities in supporting grid energy efficiency. By analyzing the boundary operating characteristics of the photovoltaic power generation system after it is integrated into the grid, the typical operation load rate and multi-spatial-temporal operation mode of reactive power service in photovoltaic power generation systems are simulated. The correlation effect analysis model of photovoltaic power system reactive power service to support grid energy efficiency support based on the Pearson method and power system voltage deviation, active loss, and line loss rate as the core indicators are designed. At the same time, a mean-semi-absolute deviation and t-test system operation energy efficiency correlation effect evaluation model is constructed. Through calculating the deviation of voltage or energy consumption, the difference in the impact of the reactive service capability of inverters or SVG devices in the photovoltaic power generation systems on changes in the energy efficiency of grid operations is verified. The mechanism of the photovoltaic system reactive power service to the energy efficiency support of grid operation is revealed. Research conclusions show that: (1) The model constructed in this paper can effectively analyze the impact and change trend of renewable energy on the system operation energy efficiency after it is integrated into the power grid. The research data can provide a decision basis for tapping the energy efficiency improvement space and has a certain reference value on optimizing the design of renewable energy grid-connected projects and investment cost control. (2) Under the combination of different operation modes of the system based on multi-spatial-temporal scales and the reactive power support of the photovoltaic power generation system, the reactive power support capacity of the photovoltaic power generation system can generally improve the operational energy efficiency of the power grid, that is, the impact on the power efficiency of the power system operation is the positively correlated effect. This conclusion provides a theoretical basis for promoting its increase in the proportion of consumption in power transactions. Research-based on the market value of reactive power services has

enriched the theoretical system of renewable energy participating in auxiliary service market transactions. (3) Taking the effect of the reactive power service of the photovoltaic power generation system on the grid energy efficiency as an example, this article reveals the mechanism of the impact of renewable energy on the grid energy efficiency after grid connection. The energy efficiency change data of the multi-spatial-temporal operation mode can provide theoretical support for the auxiliary service transaction decision in the real-time power balance market. (4) The analysis method based on the auxiliary service of the photovoltaic power generation system to support the dynamic energy efficiency of the power grid provides ideas and theoretical references for the research on the energy efficiency management decision-making of the photovoltaic cluster or the wind and solar cluster.

## Supporting information

**S1 Table. Abbreviated list of nouns.**
(DOCX)

## Author Contributions

**Conceptualization:** Cui Yong.

**Data curation:** Mingzhen Shao, Liu Wen, Thomas Stephen Ramsey.

**Formal analysis:** Mingzhen Shao, Ji Desen.

**Methodology:** Zhou Xiaoqian.

**Resources:** Cui Yong.

**Software:** Mingzhen Shao.

**Validation:** Cui Yong.

**Writing – original draft:** Cui Yong.

**Writing – review & editing:** Cui Yong.

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
