## [Decision Letter · Decision Letter 0]

26 Jan 2022

PONE-D-22-00222Research on Supporting Mechanism of Reactive Power Service of PV System to Grid Energy Efficiency Based on Multi-time and Space-time OperationPLOS ONE

Dear Dr. CUI,

Thank you for submitting your manuscript to PLOS ONE. After careful consideration, we feel that it has merit but does not fully meet PLOS ONE’s publication criteria as it currently stands. Therefore, we invite you to submit a revised version of the manuscript that addresses the points raised during the review process.

We look forward to receiving your revised manuscript.

Kind regards,

Ziqiang Zeng, Ph.D.

Academic Editor

PLOS ONE

Journal Requirements:

Additional Editor Comments:

According to the reviewers' comments, this paper needs a major revision before it can be further considered. The authors are expected to carefully revise this paper by addressing all the issues raised by the reviewers.

Reviewers' comments:

Reviewer's Responses to Questions

**Comments to the Author**

1. Is the manuscript technically sound, and do the data support the conclusions?

Reviewer #1: Yes

Reviewer #2: Yes

2. Has the statistical analysis been performed appropriately and rigorously? 

Reviewer #1: Yes

Reviewer #2: Yes

3. Have the authors made all data underlying the findings in their manuscript fully available?

Reviewer #1: Yes

Reviewer #2: Yes

4. Is the manuscript presented in an intelligible fashion and written in standard English?

Reviewer #1: Yes

Reviewer #2: Yes

5. Review Comments to the Author

Reviewer #1: The author presented a study on Mechanism of Reactive Power Service of PV System to Grid Energy Efficiency Based on Multi-time and Space-time Operation. The study is interesting, which I think merits a publication. However, I have the following comments:

1) The authors should consider harmonic issue and discusses how the proposed solution can mitigate the said problem. An extensive review of research in harmonic mitigation effort has been published in ["Advances in reduction of total harmonic distortion in solar photovoltaic systems: A literature review", International journal of energy research] and this can serve as a starting point for carrying out this addition. A research article on harmonic mitigation of PV system has also been published in ["Predictive Adaptive Filter for Reducing Total Harmonics Distortion in PV Systems", Energies]. The author can choose to either discuss the harmonic mitigation effort in a new section or perform an additional harmonic analysis.

2) The authors should benchmark their work with similar studies performed in ["A high-gain reflex-based bidirectional DC charger with efficient energy recycling for low-voltage battery charging-discharging power control", Energies], ["A new combined boost converter with improved voltage gain as a battery-powered front-end interface for automotive audio amplifiers", Energies] and ["Study of a Bidirectional Power Converter Integrated with Battery/Ultracapacitor Dual-Energy Storage", Energies].

Reviewer #2: The article presents an interesting Research on Supporting Mechanism of Reactive Power Service of PV System to Grid Energy Efficiency Based on Multi-time and Space-time Operation. the article is well written, but it may be let down at its current form due the lack of flow or presentation quality:

1- The main contribution of the article is not clearly highlighted.

2- The introduction is too long and as a reader we may loss the main point of the article. Please revise it with better flow that highlights the issue and the contribution of the research.

3- All figures quality are not acceptable especially Fig 1. Please redraw this figure and enhance the fonts quality in others.

4- Ref 2 is not cited in the text, and there is no any reference of last 3 years!!!.

5- Adding a table of abbreviation could clear many points to readers.

6. Table of comparison of conducted work and related works in literature will point out the improvement of the findings.

6. PLOS authors have the option to publish the peer review history of their article (what does this mean?). If published, this will include your full peer review and any attached files.

Reviewer #1: No

Reviewer #2: **Yes: **Dr. Mohamed Salem

---

## [Author Response · Author response to Decision Letter 0]

3 Mar 2022

The reviewer's comments have been replied and answered one by one in the “Response to Reviewers”

---

## [Decision Letter · Decision Letter 1]

25 Apr 2022

Research on Supporting Mechanism of Ancillary Service of PV System to Grid Energy Efficiency Based on Multi-time and Space-time Operation

PONE-D-22-00222R1

Dear Dr. Shao,

We’re pleased to inform you that your manuscript has been judged scientifically suitable for publication and will be formally accepted for publication once it meets all outstanding technical requirements.

Kind regards,

Ziqiang Zeng, Ph.D.

Academic Editor

PLOS ONE

Additional Editor Comments (optional):

Based on the referees' review comments, this paper could be accepted.

Reviewers' comments:

Reviewer's Responses to Questions

**Comments to the Author**

1. If the authors have adequately addressed your comments raised in a previous round of review and you feel that this manuscript is now acceptable for publication, you may indicate that here to bypass the “Comments to the Author” section, enter your conflict of interest statement in the “Confidential to Editor” section, and submit your "Accept" recommendation.

Reviewer #1: All comments have been addressed

Reviewer #3: All comments have been addressed

2. Is the manuscript technically sound, and do the data support the conclusions?

Reviewer #1: Yes

Reviewer #3: Yes

3. Has the statistical analysis been performed appropriately and rigorously? 

Reviewer #1: Yes

Reviewer #3: Yes

4. Have the authors made all data underlying the findings in their manuscript fully available?

Reviewer #1: Yes

Reviewer #3: Yes

5. Is the manuscript presented in an intelligible fashion and written in standard English?

Reviewer #1: Yes

Reviewer #3: Yes

6. Review Comments to the Author

Reviewer #1: All comments have been addressed and there is no more additional comments. Thank you for the addressing the comments.

Reviewer #3: (No Response)

7. PLOS authors have the option to publish the peer review history of their article (what does this mean?). If published, this will include your full peer review and any attached files.

Reviewer #1: No

Reviewer #3: No

---

## [Editor Report · Acceptance letter]

5 May 2022

PONE-D-22-00222R1 

Research on Supporting Mechanism of Ancillary Service of PV System to Grid Energy Efficiency Based on Multi-time and Space-time Operation 

Dear Dr. Shao:

I'm pleased to inform you that your manuscript has been deemed suitable for publication in PLOS ONE. Congratulations! Your manuscript is now with our production department. 

Kind regards, 

on behalf of

Dr. Ziqiang Zeng 

Academic Editor

PLOS ONE